# Monitoring the Geometric Position of Transition Zones to Increase the Quality and Safety of Railway Lines

**Stanislav Hodas** [1,*], **Jana Izvoltova** [2], **Jakub Chromcak** [2] **and Dasa Bacova** [2]

1   Department of Railway Engineering and Track Management, University of Zilina, Univerzitna 8215/1, 010 26 Zilina, Slovakia
2   Department of Geodesy, University of Zilina, Univerzitna 8215/1, 010 26 Zilina, Slovakia; jana.izvoltova@uniza.sk (J.I.); jakub.chromcak@uniza.sk (J.C.); dasa.bacova@uniza.sk (D.B.)
\*   Correspondence: stanislav.hodas@uniza.sk; Tel.: +421-41-513-5847

**Abstract:** Transition zones on railway lines are localities with gradual changes in the construction layers in the connections between a fixed track and ballasted track or between a wide track and various railway objects, e.g., tunnels, bridges, culverts, etc. The different type of construction of transition zones causes a shock wave when the train passes, which can cause undesired effects on the stability of its construction, durability, and passengers' comfort. For this reason, railway opera-tors pay increased attention to the construction inspection of these transition sections. The research deals with the description of the transition zones, methods, and results of their monitoring in experimental sections of the railway corridors. Innovative aspects are the measurements made using precise geodetic instruments as well as continuous measurements with the KRAB trolley. The analyses of measurements in the experimental sections of the track show whether the stability of the geometric spatial position is ensured.

**Keywords:** railway engineering; transition zones; construction modification; safety; inspection; continuous measurements

## 1. Introduction

Railway transition zones (TZ) are places with a higher probability of deformation of the geometric and structural arrangement of the track because the change in the construction of the railway superstructure and substructure will cause a different distribution of forces when changing from the "soft" to the "solid" type of structure. The behavior of the structural layers changes under different wheel pressures and dynamic forces. The solution is in designing structural modifications that distribute and eliminate these forces in a certain short section of track.

A typical example of a transition zone is the construction between ballasted and fixed tracks where the types of particular construction layers change and are supplemented by suitable subgrade, stabilization (for example cement bound materials—CBM), granular layers (unbound granular materials—UGM), geo-accessories, etc.

The railway surface creates waves in the longitudinal direction under traffic loading and the greatest effect arises under a locomotive, which pushes a sinusoidal shock wave. In the case of a homogeneous railway structure, the deflection wave has a smooth course in the longitudinal direction. The problem occurs when the wave hits a solid obstacle, which may be the different construction of the railway body. The effect of the forces depends on the type of transition structure, operating speed, the built-in materials, and the overall quality of the construction. Otherwise, defects in spatial geometrical track position will occur.

The solution to this problem is in an individual approach to the process of designing the constructional layers of the transition structure and a very precise calculation of physical influences. Inserting other constructional elements into the railway body should help to eliminate or minimize the occurrence of destructive forces acting on the transition zone,

such as various reinforced concrete slabs or tubs, reinforced subbase layers, various granular layers, geological or synthetic reinforcement, etc.

## 2. Elimination of Unwanted Shock Wave Forces

Over the last twenty years, various designs of transition zone shapes have been made in the world, which were intended to increase the stability of the railway body and its structural layers. In particular, it is a matter of ensuring the long-term stability of the spatial geometric position of the track (GPT). An important factor is the shape of the transition zone elements, which are optimized according to Figure 1, for example.

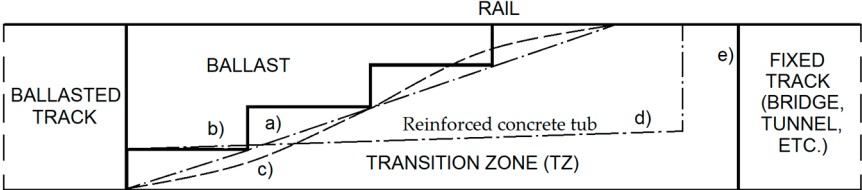

**Figure 1.** Shapes of elements for elimination of forces in the transition zones, optimized: (**a**) Linear; (**b**) Stepped; (**c**) Cosine; (**d**) Reinforced concrete tub; (**e**) Non-optimized.

On the railway line, it is possible to drive on the tracks in both directions, and therefore we recognize two driving directions that will affect the strength and direction of these effects of the shock wave. The train can run in the direction from the ballasted trackbed to the fixed track, where when reaching this point it seems to hit an obstacle, and the forces breaking the GPT are created here. The train run can also be reversed. From the fixed point, it would "jump" into the trackbed and push the structural layers and thus deform the GPT (part representing the elastic deformation of the structure as reversible and undesirable as irreversible).

The optimization of the shape of the structural layers of the railway body is usually performed by reinforced concrete slabs and tubs on the longitudinal slope. The designers design the shape and size of these concrete elements, including the base layers of the various fractions and the stabilization of the construction materials.

## 3. Troubleshooting in Transition Zones

The authors, who have recently addressed the issue, individually approach the design proposals with calculations from different perspectives. For example, an overview of the proposed building elements in some countries is provided in [1], which shows the values of technical characteristics of the materials used in each layer. In the literature, transitions from the railway bed to concrete building structures, such as bridges, are addressed. The authors summarized the summary of methods used in the design of TZ in different countries of the world, which they compared with the lines in Portugal.

Some manufacturing companies design prefabricated products for their use in TZ, such as Getzner Engineering [2], whose products are optimized, including fastening materials for track superstructure, trackbed, base layers, and calculations are performed within realistic finite element methods (FEM models).

The literature dealing with TZ includes [3–8], which solve various approaches to the design of structural layers in TZ [3], such as the transition from ballasted to fixed track in [4], ballast material and insulation layers in TZ, including gluing trackbed grains with reinforcement (see [5]). Modeling of railway track temperature regime in winter periods with real heat-technical values for different climatic characteristics was published by Hodas [6]. Research at Rohmbergk Rail [7] addressed a comprehensive design as a V-TRAS module, which allows the transition between the terminating trackbed and the connection to the solids of the fixed track or the bridge.

In [8], the designs of longitudinal profiles of TZ are solved as solutions by jumps, linear, or their cosine course, on which they verified the vertical displacements of GPT between ballasted track and bridge structures, or culverts. In [9], the research focused on the modifications and maintenance of TZ, where the rail decreases in TZ or their elevation

heights were adjusted. The action of forces and pressures from the wheels is addressed in [10], where the dynamic stress system with continuously welded rails (CWR), which arises from a moving train, was evaluated.

The transition from the embankment to the bridge objects is dealt with in [11], where there are solutions of types of trackbed to bridge, such as trackbed to fixed track. Special structural designs are proposed in [12], where CBM and UGM materials with a certain concrete chamber in front of the bridge are designed between the given structures, such as ballasted track to bridge, fixed track to bridge, etc. The authors examine in detail the stress of these elements of transition zones using computer models, which compares the behavior of structures with and without TZ.

### 3.1. Rail Bed Transition to a Fixed Track

Research tasks took place mainly in these areas of railway practice, such as numerical modeling of transit zones and comparison of models of these transit zones with "in-situ" heavy laboratory models built at the university. The main focus of the presented research in the paper are geodetic measurements of the spatial position of track axes in transit zones from the point of view of surveyors and continuous measurements on corridor lines by measuring trolleys from the point of view of railway engineers.

A collision of fixed-type railway structures occurs mainly on a wider line, or at railway stations. The proposed building element mitigates the undesirable effects of the shock wave of the train and will have the shape of a reinforced concrete slab (such as a concrete tub for ballast) for the railway line according to Figure 2. The proposed lengths of $l_{TZ}$ for individual speeds are calculated in Table 1, including the final concrete block in the required form for terminating the fixed track.

**Table 1.** Design lengths of transition zones between fixed and ballasted tracks.

| $V$ (km/h) | Length $l_{TZ}$ (m) |
| --- | --- |
| 120 | 16.67 |
| 140 | 19.45 |
| 160 | 22.22 [1] |
| 200 | 27.78 |
| 250 | 34.73 |
| 300 | 41.67 |

[1] Figure 2 at $V$ = 160 km/h.

The structural layers of the transition zone with built-in building elements and base layers are proposed in Figure 2 (corridor tunnel in Trencianske Bohuslavice) with a longitudinal slope of the reinforced concrete slab (tub with ballast), where the thickness of the trackbed changes from 250 to 350 mm below the storage area of the basement [13].

### 3.2. Ballasted Trackbed and Bridge Structures

For bridge structures, the establishment of transition zones is also important from a design point of view, especially at high speeds of already $V \geq 160$ km/h, not only on the line type of ballast-fixed track but for all buildings that form a certain type of obstacle that reduces the effects of the shock wave in the railway body (Figure 3a). Deformations of the spatial geometric position of the track (GPT) can occur during the shock wave impact (Figure 3b) [2].

These objects need to be protected by a building intermediate stage between two different structures of the railway body (ballasted–fixed track) by inserting a reinforced concrete slab or by modifying the material of the railway top and bottom, for example, according to Figure 4. The reinforced concrete slab is placed on the support protrusion already created during the construction of the bridge or it is concreted at the support. Care should be taken behind the supports of bridges and culverts as there is a risk of a significant drop in the layers of material due to poor compaction. The new transition zone cannot load

the bridge object with additional forces and also transmit the effects of the shock wave, as deformations of the bridge object could occur.

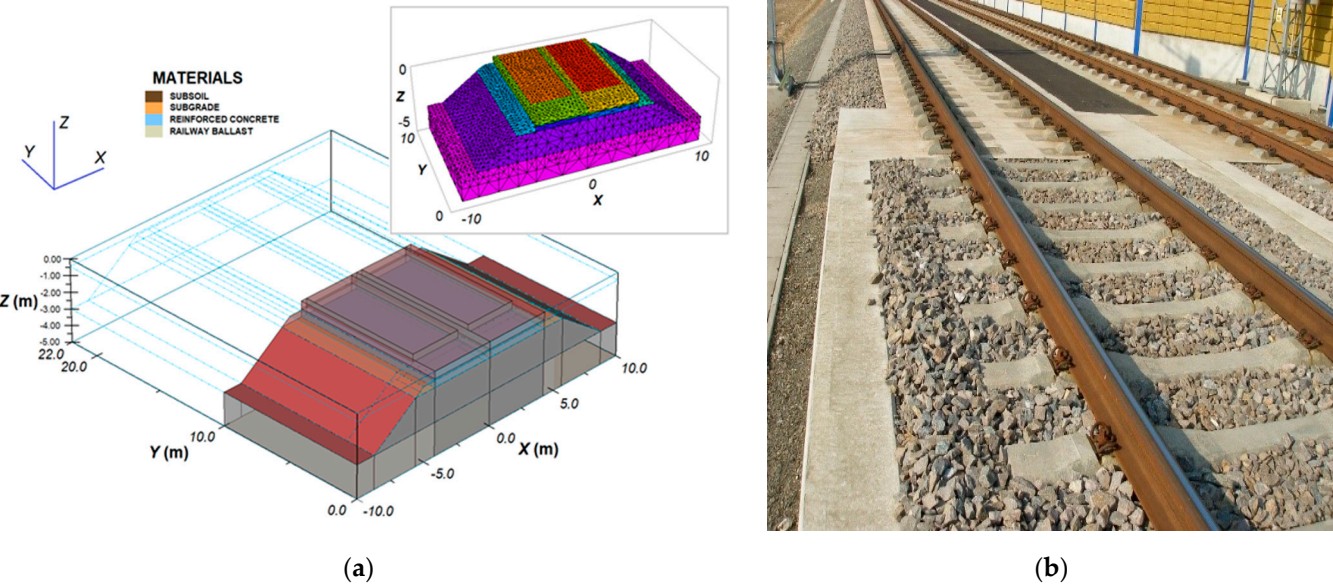

(**a**)  (**b**)

**Figure 2.** Transition zone near the tunnel Turecky Vrch in Trencianske Bohuslavice: ballasted and fixed track with linear reinforced concrete tub, $V$ = 160 km/h: (**a**) Numerical design—structural elements [6,14]; (**b**) Photo: Reinforced concrete tub with ballast on railway main corridor [Hodas].

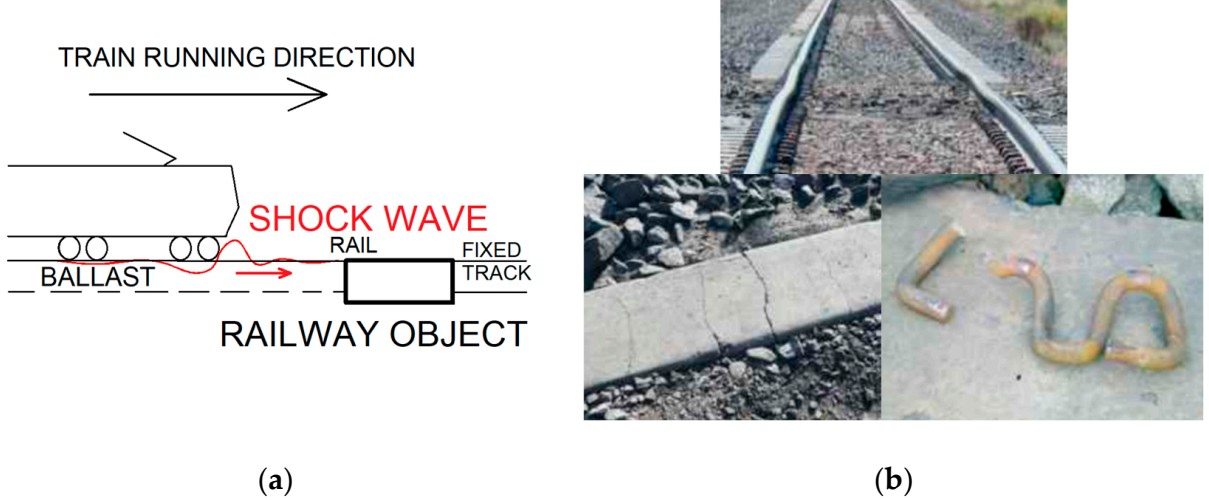

(**a**)  (**b**)

**Figure 3.** Shock wave caused by a train running: (**a**) Shock wave on all fixed railway objects [13]; (**b**) Photo: Deformation of geometric position and heights [2].

There is also a design of the TZ type of ballasted and fixed tracks with a reinforced concrete tub, but this would be demanding, as there are a large number of bridge structures, culverts, etc. on the railway line, which are close in height to the track gradient. Transitional reinforced concrete tubs are inserted mainly if the ballasted track changes to the fixed track of the bridge object [11].

### 3.3. Tunnel Objects with a Fixed Track

The transition zones of the tunnel portals are particularly important (if there is a conflict between the ballasted and fixed track too) [13,15], as it is necessary to ensure increased security of the train entry into the interior of a tunnel track. The TZ will be placed in front of the tunnel portals and will also be used as an entry for rescue teams into the

tunnel tube (in case of rescue or maintenance works). If there is a trackbed (ballast) in front of the tunnel, a transition zone will be built as a reinforced concrete tub (Figure 5) to eliminate the shock wave on the portals. If there is also a fixed track in front of the tunnel, the transition reinforced concrete tub will be built further from the tunnel portal in this ballast to the fixed structural layer change transition (then there is no impact on the tunnel), but the fixed track and tunnel transition must be completed in front of its portals.

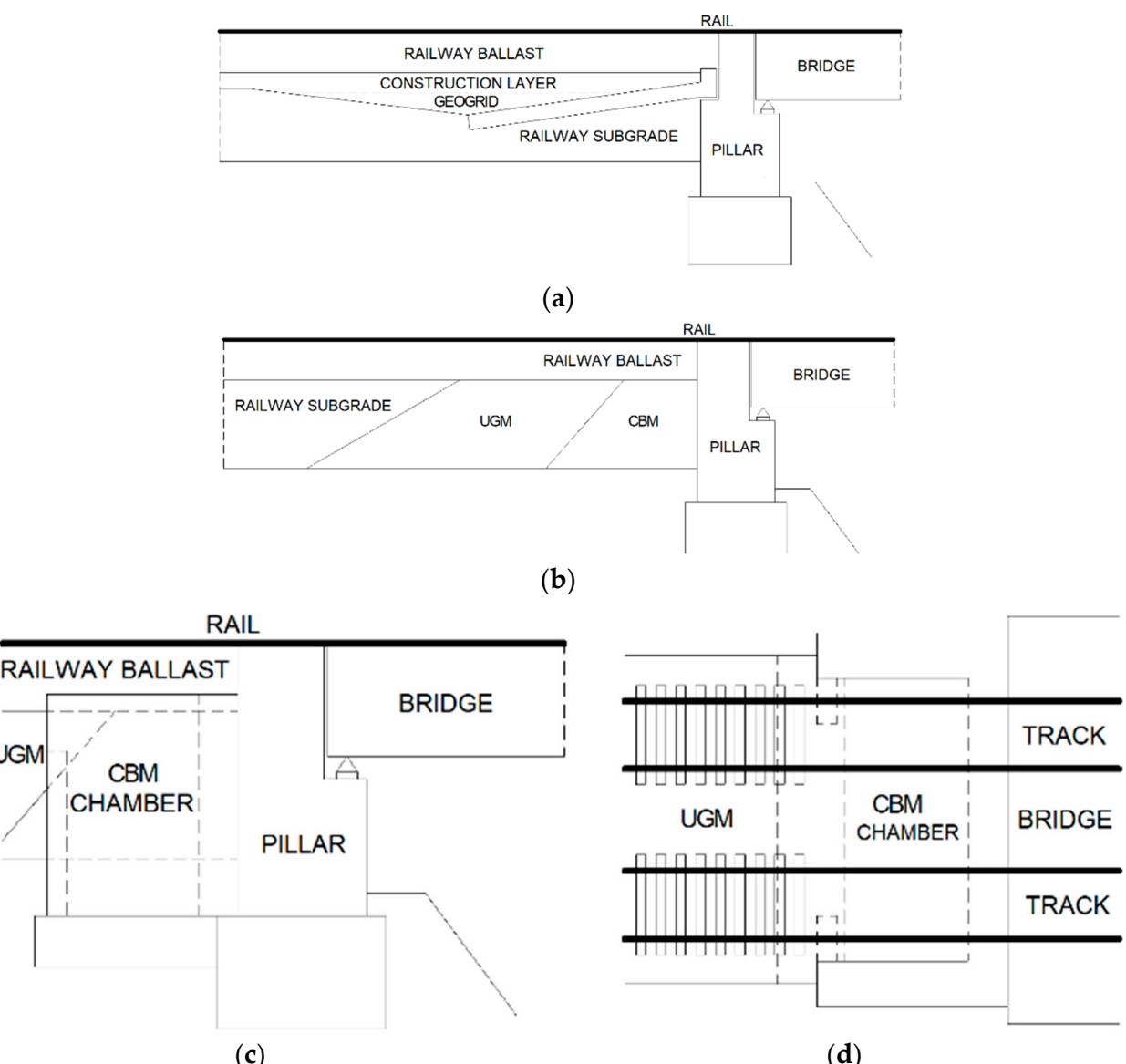

**Figure 4.** Design of transition zones as ballasted track and bridge: (**a**) Reinforced concrete slab; (**b**) CBM and UGM material; with CBM chamber: (**c**) Section view and (**d**) Plan view.

In railway practice, there are usually several construction objects in close succession, for example, a line led in a tunnel passes to a bridge object (Figure 5c), etc., and there is usually a line formed by a fixed or ballasted track between the objects. Railway tunnels are generally designed for higher line speeds so that they can be used to increase speeds in future line modernizations and the speed in the tunnel will not be reduced in the future. An example is the built double-track railway tunnel in Figure 5b,c in Trencianske Bohuslavice, which is designed for the speed of $V = 200$ km/h, whereas the modernized main European *Va* corridor Bratislava–Zilina is currently driven at $V = 160$ km/h.

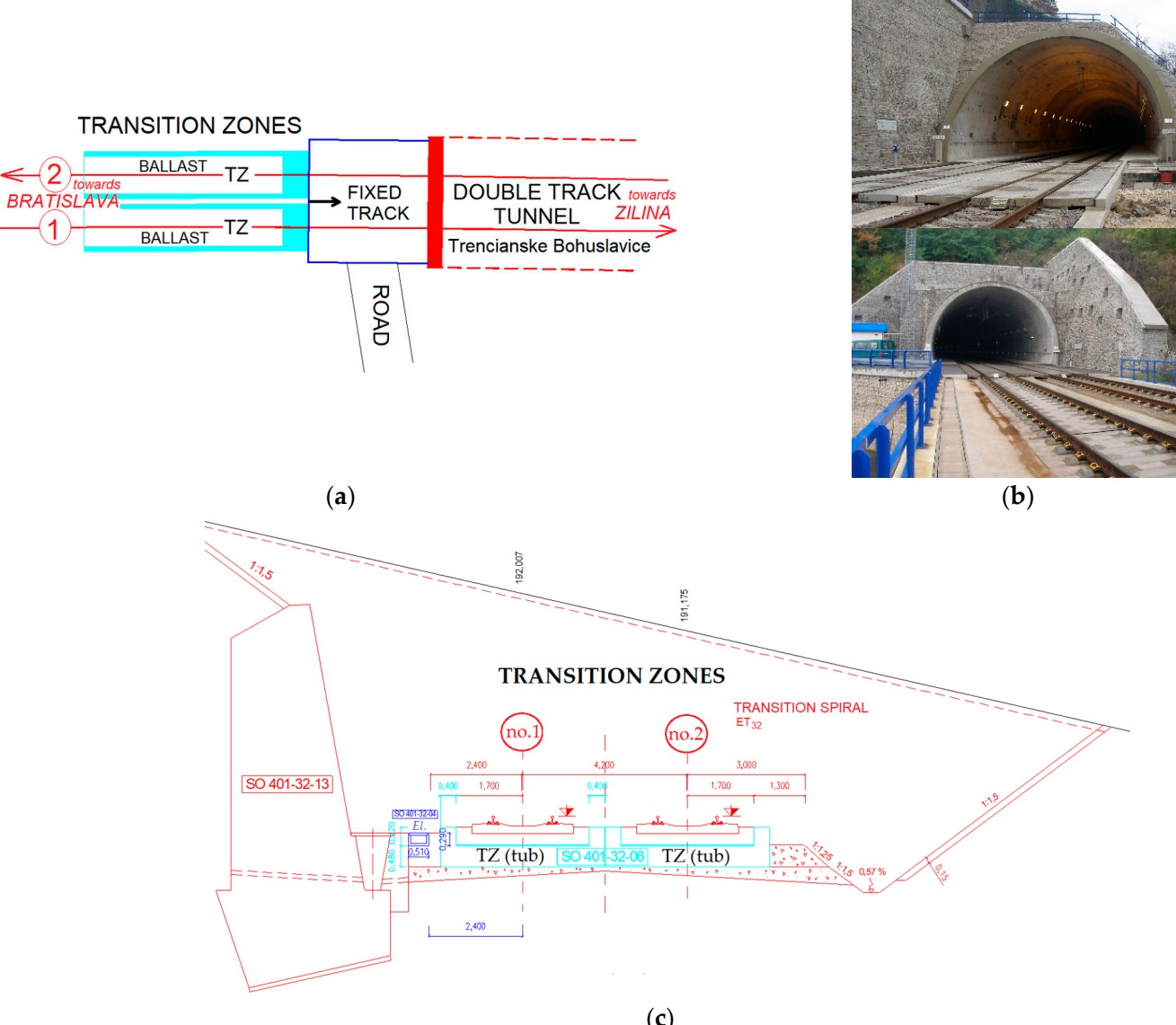

**Figure 5.** Railway tunnel Trencianske Bohuslavice *V* = 200 km/h [13]: (**a**) Situation of TZ; (**b**) Photo: TZ in front of the tunnel and bridge with a fixed track [Hodas]; (**c**) Designed cross-section with TZ in [15].

## 4. Diagnostics of Transition Zones in the Experimental Section by Geodetic Methods

Since 2017, six geodetic measurements of transition zones, at the ends of the fixed tracks, have been carried out in the locality "Trencin—A new railway bridge" in Figure 6, [16]. The measurements of TZ took place in the section towards "Bratislava" on the right side of the river Váh in 122.219–122.239 km (both track axes) and in the section towards "Zilina" on the left side in 122.748–122.768 km in track axis no. 1 and 122.763–122.783 km in track axis no. 2. Height measurements were performed on both track axes (no. 1 and no. 2) and both their rails, with detailed points being signaled in the longitudinal direction of the track at a distance of about 5 m from each other, which represents every eighth fastening of the rail to the sleeper. Measurements were performed with a DNA03 digital leveling device with a unit standard deviation $m_0$ = 0.3 mm, which represents the accuracy of determining the elevation of a 1 km long leveling. To maintain the accuracy and unambiguity of the results, when comparing the individual measuring stages, the calculated height changes of the track were determined separately for the right and left rails (with cant *D* of the track curve).

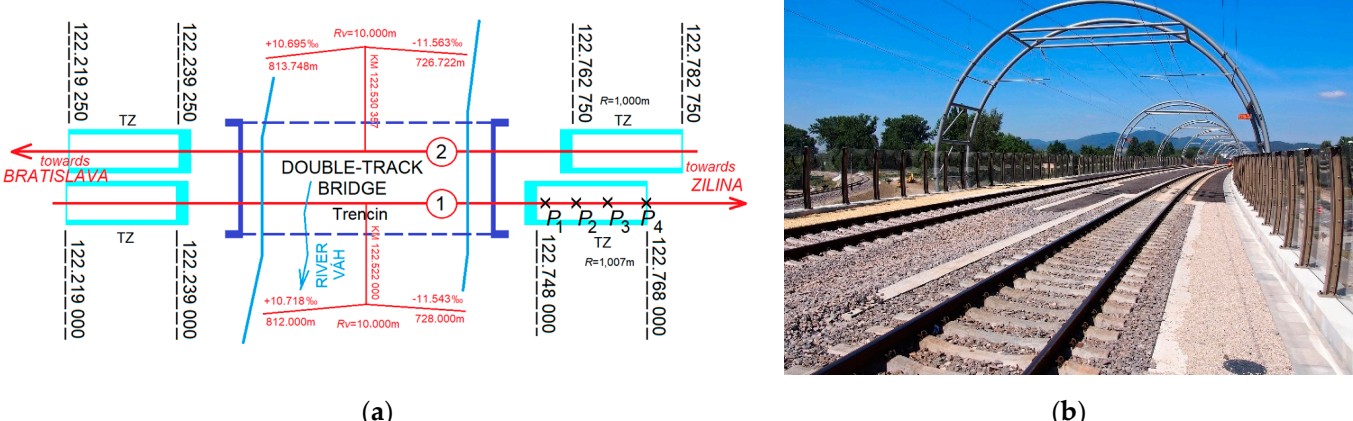

(**a**)                                                                                                    (**b**)

**Figure 6.** Railway double-track bridge in Trencin $V$ = 140 km/h: (**a**) Scheme of TZ; (**b**) Photo: Transition zones—ballasted track to bridge with a fixed track [13,16].

At each stage, the geodetic measurements are related to the reference height network, which consists of permanently stabilized marks, stabilized either at the base of the masts or in the concrete curb of the railway bridge. The heights of the reference points were determined by precise leveling.

Evaluation of height changes of transition zones. The most critical parts of the construction of a fixed track for the height changes of the track, affected by the traffic load, are the transition zones, i. j. between a fixed track and the classic construction of a railway superstructure with a ballast bed. In the transition zone in the part towards "Bratislava" in KM-position in Figure 6a, four points $P_i$ in track axes no. 1 and no. 2 are measured, the final height changes of which are shown by the graphs of time dependence in Figure 7a,b.

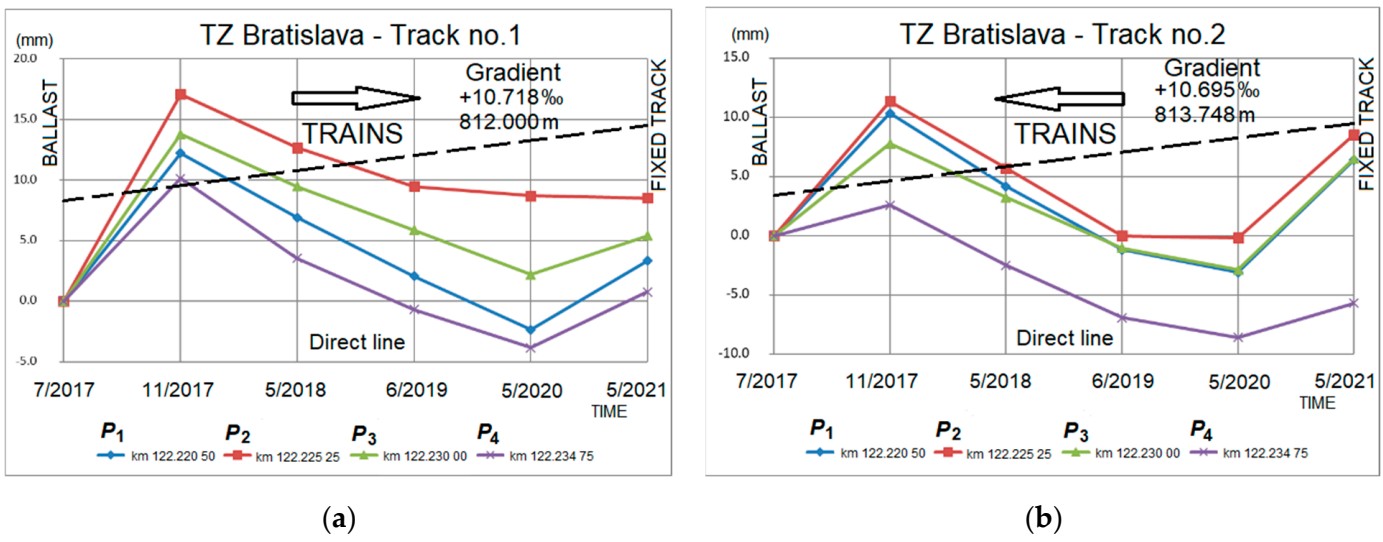

(**a**)                                                                                                    (**b**)

**Figure 7.** Time dependence of height changes of TZ, towards "Bratislava": (**a**) Track axis no. 1; (**b**) Track axis no. 2.

The transition zone in the part of the track towards "Zilina" are in Figure 6, and the height changes are shown by the time dependence graphs in Figure 8. Their relative position of TZ is shifted in the longitudinal direction in the photo in Figure 6b.

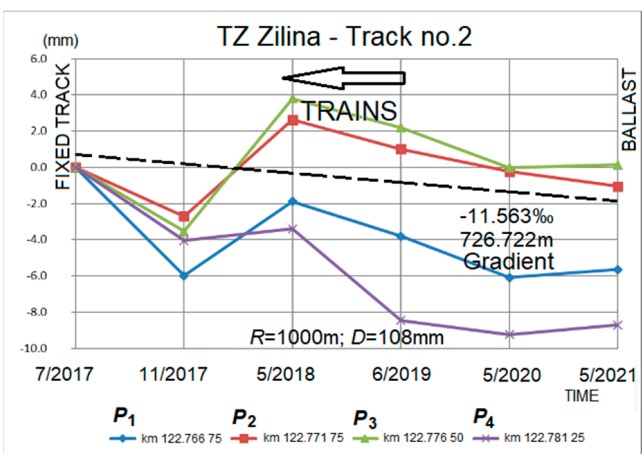

**Figure 8.** Time dependence of height changes of TZ, towards "Zilina": Track axis no. 2.

## 5. Continuous Monitoring and Analysis of Track Geometry Quality in Transition Zones

The main task of VEGA research [13] at the Department of Railway Engineering is to monitor the behavior of the spatial geometric position of the track axes on several experimental sections on the main railway corridor Bratislava-Zilina-Kosice in the Slovak Republic (within the European Corridor *Va*). The first sections of TZ are located within the construction of the double-track tunnel "Turecky Vrch" in Trencianske Bohuslavice city in Figures 2b and 5b (measurements 2012–2022), and the second experimental area is located on the construction of a double-track railway bridge in Trencin in Figure 6b (measurements 2017–2022). The purpose of monitoring is to measure the spatial geometric position of the track axes, immediately after the incorporation of building elements into the transition zones. Experimental measurements of these transit zones were performed continuously by a KRAB measuring trolley [17] with a step of 250 mm with wavebands $\lambda = 1 \div 25$ m in Figure 9 within a complete section of the track.

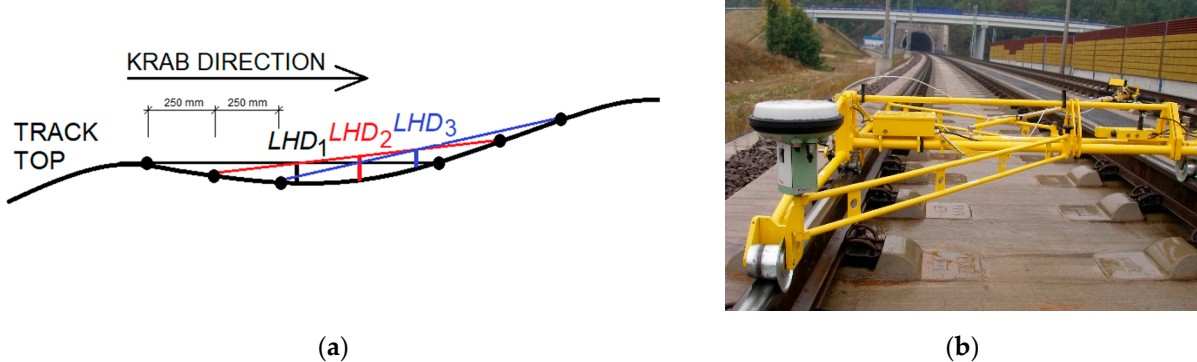

(**a**)                                                                 (**b**)

**Figure 9.** Monitoring the quality of track geometry—by KRAB trolley measurements [13]: (**a**) Principle of continuous measurements of height changes LHD; (**b**) Photo [Hodas].

Operating deviations of geometric quantities in the evaluation of measurement results are defined in three stages; $V = 160$ km/h are shown in Table 2 [18]:

- degree AL—Alert Limit: if the value is exceeded, it is necessary to assess the detected change and plan future adjustments,
- degree IL—Intervention Limit (Corrections): if the value is exceeded, maintenance works are performed,
- degree IAL—Immediate Action Limit: measures need to be taken to ensure that the risk is reduced to an acceptable level.

**Table 2.** Operating deviations AL, IL, and IAL [18] at speed of *V* = 160 km/h.

| Parameters of Inspection | | Operating Deviations | | Marginal Operating Deviations |
|---|---|---|---|---|
| | | AL (mm) | IL (mm) | IAL (mm) |
| TG | Track gauge | +15/−4 | +20/−6 | +25/−7 |
| RD | Rail deformation | - | ±4 | ±5 |
| LHD | Longitudinal height deviations [1] | ±7 | ±10 | ±17 |
| TD | Track direction | ±6 | ±8 | ±10 |
| TC | Track cant | ±5 | ±6 | ±8 |

[1] Figure 9a for the longitudinal height deviations (LHD).

The paper presents the results of monitoring the course of heights behavior in track axes in transition zones (longitudinal height deviations—LHD) in our experiment for speed on railway corridors in the Slovak Republic *V* = 160 km/h. Other monitoring parameters (track gauge—TG, rail deformation—RD, track direction—TD, and track cant of rails—TC) were also monitored by the KRAB measuring trolley according to the [18] standard but are not the subject of this text. During the measuring run, the following so-called primary track values are [17]:

- gauge (potentiometer transducer on the left wheel),
- alignment (lateral versine) of the right rail,
- top (vertical versine) of the right rail,
- cant (new, high reliable, and precise inclinometer),
- track gradient (option),
- track distance (odometer-optical encoder),
- measuring speed.

The following items of geometry inspection were available:

- actual alignment and level in wavebands $\lambda = 1 \div 25$ m,
- separation of all geometric signals into long wave ($\lambda > 25$ m) and short wave ($\lambda < 25$ m) parts,
- so-called section evaluation—statistic evaluation of the track geometry based on standard deviation and quality index,
- table of local defects, print out of geometrical lay, and tables.

At the end of the evaluation of the measurements [13], we can state that the transition zones, using the proposed building elements, represent an increase in the quality and sustainability of the spatial geometry of the track, which is proven by Figures 10 and 11. TZ objects are built in places where it is necessary to reduce the effects of the shock wave, which is caused by trains running in contact with fixed obstacles (fixed track, bridge, tunnel, culvert, etc.).

The results of measurements in the experimental track sections show that the stability of the geometric position is ensured (in our case the vertical alignment). According to the results of the inspection, it is proven that all the parameters of the limit deviations in Table 2 were met (for a multi-year period). Reasons for material drops and elevations of the railway body material in the transition zones are described in detail in the analysis of measurements, where the resulting height shifts of the track axes in the TZ are also located (in contrast to this continuous monitoring by KRAB). For the first time, the tracks were tamped before the facilities were put into operation during the period of August 2017 (new buildings). Due to the confirmed stability of the geometric position and height, during the monitoring by continuous measurement with the measuring trolley KRAB [17], the second tamping of the tracks took place in November 2021, depending on the criteria in Table 2 (the tamping works had to be done after 4 years).

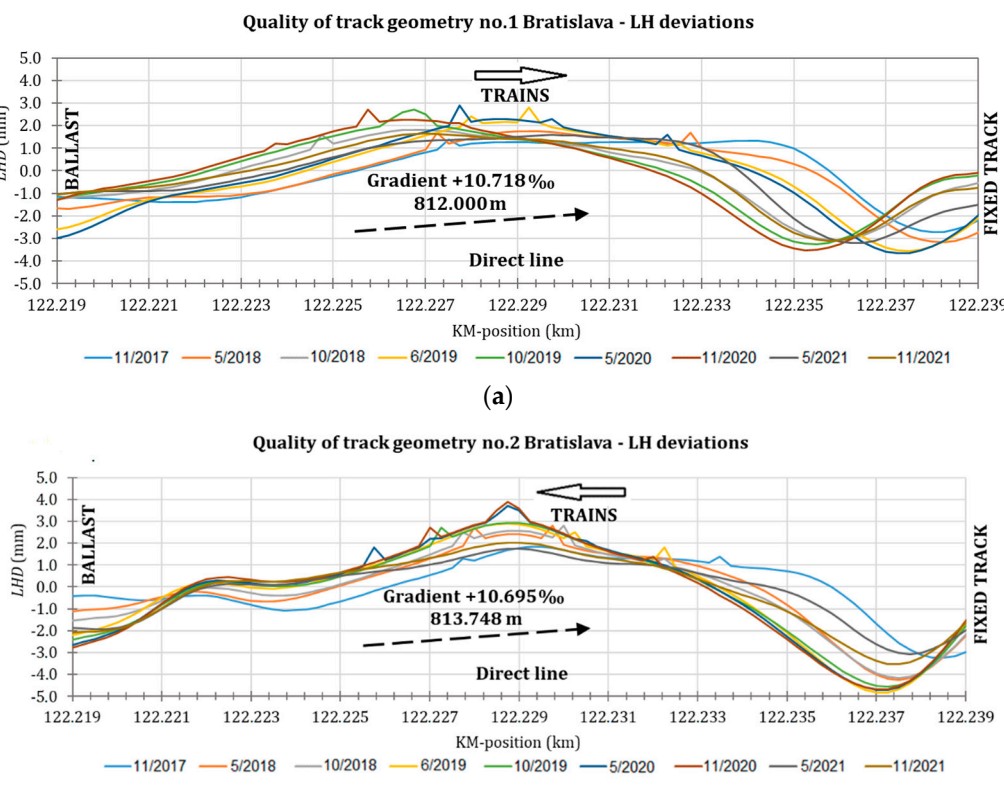

**Figure 10.** Quality of track geometry—longitudinal height deviations (LHD) of the transition zone, toward "Bratislava": (**a**) Track axis no. 1; (**b**) Track axis no. 2.

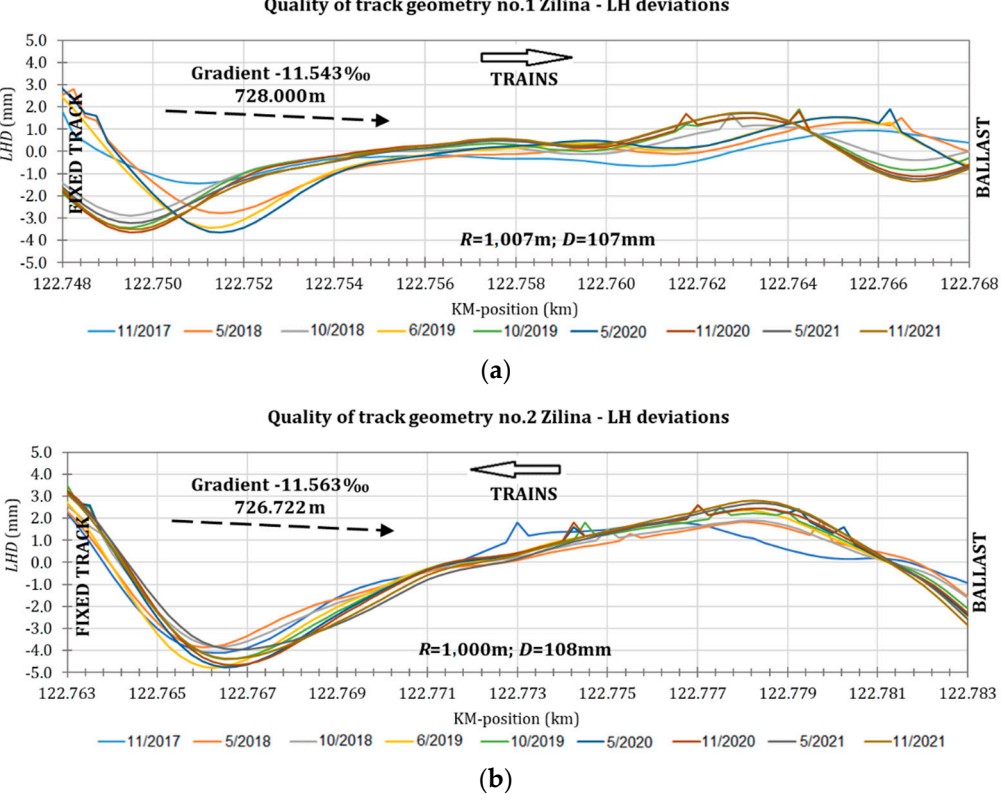

**Figure 11.** Quality of track geometry—longitudinal height deviations (LHD) of the transition zone, toward "Zilina": (**a**) Track axis no. 1; (**b**) Track axis no. 2.

The height shifts of the particular points $P_i$ by geodetic measurements are always related to the first point depending on the KM-positioning of the axis of the track. The predominant direction of travel of the trainsets (approx. 90%) is shown in Figures 7–11, using the arrow sign "→ TRAINS". All objects of the transition zones are designed in the longitudinal slopes of the gradient, and the part towards "Zilina" is located in the arches. In track axis no. 1 is radius $R_1 = 1007$ m with cant $D_1 = 107$ mm, and in track axis no. 2 is $R_2 = 1000$ m with cant $D_2 = 108$ mm. The height shifts of the tracks are mainly influenced by the predominant direction of the train's operation and their location in the curves with the cant (Figure 6a).

The graph in Figures 7a and 10a show that the trains push the gravel bed against a fixed track (concrete block at the end of TZ) in the direction of the rising gradient in the section with +10.718‰. Conversely, in Figures 7b and 10b, we can see that the train "jumps" from a fixed track into a reinforced concrete tub with a gravel bed, pushing out the material. The track gradient of axis no. 2 in this section decreases with a longitudinal slope of −10.695‰ in the opposite direction of train travel.

Figure 10a shows that the train "jumps" from the concrete block of the fixed track to the tub with the ballast of TZ in the descending direction of the track level at a gradient of −11.543‰. From Figures 8b and 11b, it is clear that if the train in a given section is running on a rising level slope, it pushes the material of the trackbed against the concrete block of the fixed track.

All these monitored points behave depending on the shock wave in Figure 3a. It is, of course, that the gradient of the tracks in the whole section, not only in the section of TZ, is adjusted to the designed heights regularly according to the required criteria of the STN 73 6360-2 [18], and depending on the inspection results.

## 6. Conclusions

Problem statement. The designed structural modifications of the objects in the transition zones significantly improve the conditions for maintaining the spatial geometric position of the track axes in the correct designed position and heights [15,18]. Despite these measures, the construction will not avoid the height changes of the track level, which is necessary after non-compliance with the criteria in Table 2 to adjust to the designed heights using tamping machines.

Case study description. During the inspection of the tracks in the mentioned sections, it is obvious that during the operation on the track, the heights in a certain part of the transition zone change by compressing the material and also by pushing it with impact force. As a rule, the extrusion of the material in the direction of the sinusoidal curve takes place behind the concrete block of the fixed track at each of the transition zones, as we can see in Figures 7–11. The shock wave represents a large destructive force arising from the wheels of heavy train traffic, and also, the higher the train speed, the greater the force.

The space position diagnostics and inspection of the section near the new bridge in Trencin started with the first measurement before the line was put into operation in August 2017. The subsequent further measurement took place during operation in November 2017, and all these measurements are still taking place today.

Main results. These sections must be adjusted by tamping throughout the operation so that deformation does not occur in the transition zone with the destruction of railway and building material. With a well-built transition zone, it is not necessary to adjust the track axes frequently and repeatedly, as research has shown.

**Author Contributions:** Conceptualization, S.H. and J.I.; methodology, S.H.; software, S.H.; validation, S.H., J.I. and J.C.; formal analysis, S.H., J.I. and D.B.; investigation, J.C.; resources, S.H.; data curation, S.H., J.I. and D.B.; writing—original draft preparation, S.H., J.I. and J.C.; writing—review and editing, S.H. and J.I; visualization, S.H.; supervision, S.H. and J.I; project administration, S.H.; funding acquisition, S.H. All authors have read and agreed to the published version of the manuscript.

**Funding:** This research received no external funding.

**Institutional Review Board Statement:** Not applicable.

**Informed Consent Statement:** Not applicable.

**Data Availability Statement:** Not applicable.

**Acknowledgments:** The presented parts of the paper were created within the framework of the research activities VEGA 1/0084/20 [13] by the Department of Railway Engineering and VEGA 1/0643/21 [16] by the Department of Geodesy at the Faculty of Civil Engineering of the University of Zilina (FCE-UNIZA).

**Conflicts of Interest:** The authors declare no conflict of interest.

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
