# Peer review of "Monitoring the Geometric Position of Transition Zones to Increase the Quality and Safety of Railway Lines"

_applsci, doi:10.3390/app12126038_

Round 1

Reviewer 1 Report

This is interesting and timely work. Some comments:

1. The abstract is too general, no technical information was given. Please include the methods descriptions and essential results.

2. Introduction Line 44-45: how the effect of these disposal methods and their drawbacks?

3. The structure is not easy to follow. Chapter 1 is the introduction, chapter 2 is Elimination of unwanted shock wave forces, chapter 3 is Troubleshooting in transition zones, chapter 4 is Diagnostics of transition zones in the experimental section by geodetic methods, and chapter 5 is Continuous monitoring of track geometry quality in transition zones. According to my knowledge, chapters 2 to 6 are not paralleled, and such a catalog is not explained in introduction. I would suggest adding a chapter on the research approach and flow chart.

4. The discussion chapter can be more detailed.

5. This paper is more like a literature review paper instead of an article.

Author Response

Response to Reviewer 1 Comments

Point 1: The abstract is too general, no technical information was given. Please include the methods descriptions and essential results.

Response 1: Accepted. The text of the abstract on lines 14-19 has been changed by adding research information.

Point 2: Introduction Line 44-45: how the effect of these disposal methods and their drawbacks?

Response 2: This cannot be supplemented, it is described in detail in the following chapters (see point 3).

Point 3: The structure is not easy to follow. Chapter 1 is the introduction, chapter 2 is Elimination of unwanted shock wave forces, chapter 3 is Troubleshooting in transition zones, chapter 4 is Diagnostics of transition zones in the experimental section by geodetic methods, and chapter 5 is Continuous monitoring of track geometry quality in transition zones. According to my knowledge, chapters 2 to 6 are not paralleled, and such a catalog is not explained in introduction. I would suggest adding a chapter on the research approach and flow chart.

Response 3: Accepted. The introduction of the paper was developed in detail in the following chapters (chapters 2 and 3). According to the opponent's proposal, chapters 5 and 6 were joined, as they belong together. The renumbered Chapter 6 “Conclusion” has been supplemented with new comments such as the problem statement, the aim of the study, and main results comments.

Point 4: The discussion chapter can be more detailed.

Response 4: Accepted. Chapter 6 “The analysis and discussion” was assigned to Chapter 5, to which it belongs. The new title of Chapter 5 is "Continuous monitoring and analysis of track geometry quality in transition zones". Chapter 5 discusses and analyzes the research topic.

Point 5: This paper is more like a literature review paper instead of an article.

Response 5: Explanation. The literature dealing with transition zones in the world is listed only on lines 71-104 (in Chapter 2 "Troubleshooting in transition zones"). From line 106 to the end, this is a research of our departments at the Faculty of Civil Engineering (Department of Railway Engineering and Department of Geodesy). An explanatory text has been added to line 106.

Reviewer 2 Report

The layout and structure of the publication is quite good, along with the references. However, authors could introduce some references regarding parameters Tg, RD, TD, TC and explain better table 2.

Additionally, authors could reduce a little bit the images and focus more on the analysis.

Overall, the research contribution through this work is adequate but it has to be clearly pointed out in the manuscript (i.e. contribution to knowledge of this research performed). Please state this clearly in the manuscript (possibly at the start and in the conclusions too).

Author Response

Response to Reviewer 2 Comments

Point 1: The layout and structure of the publication is quite good, along with the references. However, authors could introduce some references regarding parameters Tg, RD, TD, TC and explain better table 2.

Response 1: Accepted. The variables description TG, RD, TD, and TC were added to lines 242-243. Research and this paper mainly deal with LHD deviations only.

Point 2: Additionally, authors could reduce a little bit the images and focus more on the analysis.

Response 2: The pictures are aimed at acquainting readers with the given topic. They are not only researchable but also from railway practice because the construction of these buildings is very demanding. In part for analysis, we have a number of images that usually characterize the phenomenon. For example, characteristic images are selected from the analysis, for example, according to the direction of travel of the trainsets, etc.

Point 3: Overall, the research contribution through this work is adequate but it has to be clearly pointed out in the manuscript (i.e. contribution to knowledge of this research performed). Please state this clearly in the manuscript (possibly at the start and in the conclusions too).

Response 3: Accepted. We have prepared a few words and sentences for the abstract, introduction, and conclusions of the paper (problem statement, case study description, main results, etc.).

Reviewer 3 Report

The comments are included in the appendix

Author Response

Response to Reviewer 3 Comments

Point 1: Historically, transition curves began to be introduced in the case of the transition from straight track to curve, especially for curves with small radii and aspirations to travel at higher speeds. The task of such transition curves was and is to reduce the dynamic interactions in the vehicle-track system and to ensure the stability of the vehicle motion and ride comfort for the passenger.

Response 1: The paper deals with the height changes of objects in the transition zones, as this parameter is most influenced by the dynamics of train running. The running of trains in curves and transition spirals can affect the dynamics of the movement and forces acting on these structures of the transition zones from the wheels of the trainsets. These facts also disrupt the spatial position of the track geometry, but it is not the subject of the paper. The research analyzes the resulting acting force of all these forces as the spatial deformation of the track axes.

Point 2: The authors of the work dealt with an issue other than the role of transition curves described above. Namely, they dealt with a track section connecting two different track structures, for example a combination of a classic ballast track with a slab-free track based on a concrete slab. However, other meanings and the use of the transition curves should also be discussed in the introduction. It also requires supplementing the literature.

Response 2: The paper deals with transition zones, such as buildings to dampen the effects of changing the cross-sectional profile of a conventional ballast track to a fixed track (using buildings as transition zones to reduce the effects of a shock wave from train wheels).

Point 3: Three criteria for assessing the selection of transition curves are well known: geometric, geodetic, and dynamic. The authors of the work referred only to the monitoring of the transition curve state, taking into account only the track geometry in the transition curve zone.

Response 3: Geodetic and dynamic parameters are in standard STN 73 6360 and EN STN 138 03. These forces act on the geometric position of the track too, but this is more about maintaining the spatial position of the track on the reinforced concrete block (fixed track) of the transition zone and the transition to the conventional track (transition as a "jump" to the ballasted track or vice versa "climb" on the concrete block "from the ballasted track).

Point 4: What about the other criteria for assessing the state of the transition curves on the Bratislava - Zilina - Kosice section? The forces acting on the vehicle that occur in the transition curves are important. Did the monitoring also cover the forces? This requires a comment.

Response 4: The criteria for assessing the condition of transition spirals for the sections of the main corridor “Va“ Bratislava - Žilina - Košice are given in STN 73 6360-2 and in Chapter 5 in Table 2, where there are criteria for this speed limit of RP4 (in English SL4) for V = 160 km/h as TG, RD, LHD, TD, and TC. The forces acting on the wheels of the vehicles are given on the rail as well as the axle load for which the line is designed by railway engineers. The individual forces have not been investigated. The research analyzes the spatial axis of the track in operation on the transition zone as a result of all acting forces.

Reviewer 4 Report

Dear Authors, 

congratulations for the work done.

The study deals with transition zones related to railway lines. Common solutions to solve the railway transition zone-related problems are analysed. Results related to an experimental investigation (carried out in the period 2017-2021) on railway corridors (assessed using a geodetic device and a KRAB trolley) are presented and discussed.

In my opinion, the paper is interesting. Nevertheless, in order to improve its readability, comments (which are divided in "main considerations", and "tips") are reported in the following.

Main considerations:

1) Abstract:  The aims of the study are not clear. With the abstract in the present form, it seems that the paper is a review of the literature. If this is true, please state that in the abstract section. On the contrary, please, add more details, about the innovative aspects of the study (e.g., the measuraments carried out using geodetic device and the KRAB trolley on the experimental section) and the related results, to this section. 

2) Often, too long sentences are used in the manuscript. Please, divide them in more parts and/or use in a better way punctuation, and/or brackets and/or numbered lists.

3) Introduction: in this section of the paper, a summary of the remaining parts of the paper is needed. A simple scheme, which shows the framework of the study (e.g., "Problem statement"--> "Literature review on possible solutions"-->"Case study presentation"-->"Methods applied"-->"Results"), would be very useful.

4) Sections 6 and 7: In my opinion, these two sections should be joined together, and the main results of the paper (i.e., the reasons why the material is pushed out or against the concrete block) should be well expained and highlighted in the figures above (or using new figures).

Consequently, conclusions (which must include: i) problem statement; ii) aim of the study; iii) case study description; iv) main results) need to be re-written.  

5) A comparison between results from "static measurements" (i.e., using the geodetic device) and "dynamic measurements" (i.e., using the KRAB trolley) should be emphasised. For example, pros and cons of each approach can be added, and/or the reasons why these two approaches were used in this study should be stated. 

Tips:

Lines: 24-26: This sentence is too long. Please, divide it in more parts.

Line 31: At least a reference is needed here.

Line 45: At least a reference is needed here.

Line 48: Please, double check the term "i. j.".

Figure 4: It is better to divide Figure 4.c in two parts called, e.g., "(c) section view" and "(d) in plant view".

Line 143: At least a reference is needed here.

Best regards.

Author Response

Response to Reviewer 4 Comments

Point 1: Abstract:  The aims of the study are not clear. With the abstract in the present form, it seems that the paper is a review of the literature. If this is true, please state that in the abstract section. On the contrary, please, add more details, about the innovative aspects of the study (e.g., the measurements carried out using geodetic device and the KRAB trolley on the experimental section) and the related results, to this section.

Response 1: Accepted. The text of the abstract on lines 14-19 has been changed by adding this research information.

Point 2: Often, too long sentences are used in the manuscript. Please, divide them in more parts and/or use in a better way punctuation, and/or brackets and/or numbered lists.

Response 2: Accepted. Several sentences were divided into two parts (for example lines 24-26, 50-53, etc.). Some sentences could not be divided because they would lose their meaning.

Point 3: Introduction: in this section of the paper, a summary of the remaining parts of the paper is needed. A simple scheme, which shows the framework of the study (e.g., "Problem statement"--> "Literature review on possible solutions"-->"Case study presentation"-->"Methods applied"-->"Results"), would be very useful.

Response 3: The structure of the paper has been designed so that the reviewer's points can be explained in detail in Chapters 1-3. The problem statement is in Chapter 2, the Case study presentation in Chapters 3.1 to 3.3, Methods applied in Chapters 4 and 5, and Results in Chapters 5 and 6.

Point 4: Sections 6 and 7: In my opinion, these two sections should be joined together, and the main results of the paper (i.e., the reasons why the material is pushed out or against the concrete block) should be well explained and highlighted in the figures above (or using new figures).

Consequently, conclusions (which must include: i) problem statement; ii) aim of the study; iii) case study description; iv) main results) need to be re-written.

Response 4: Accepted. Chapter 6 “The analysis and discussion” was assigned to Chapter 5, to which it belongs. The new title of Chapter 5 is "Continuous monitoring and analysis of track geometry quality in transition zones". Chapter 5 discusses and analyzes the main goals of the research topic. In the conclusions, new parts were accepted, such as the problem statement, the aim of the study, main results, etc. in lines cca. 312 to 340.

Point 5: A comparison between results from "static measurements" (i.e., using the geodetic device) and "dynamic measurements" (i.e., using the KRAB trolley) should be emphasised. For example, pros and cons of each approach can be added, and/or the reasons why these two approaches were used in this study should be stated.

Response 5: Accepted. New texts were inserted in the abstract and conclusions of the paper (lines 14-19, 330-340, etc.).

Point 6: Tips:

Lines: 24-26: This sentence is too long. Please, divide it in more parts.

Line 31: At least a reference is needed here.

Line 45: At least a reference is needed here.

Line 48: Please, double check the term "i. j.".

Figure 4: It is better to divide Figure 4.c in two parts called, e.g., "(c) section view" and "(d) in plant view".

Line 143: At least a reference is needed here.

Response 6:

Lines: 24-26: Accepted.

Line 31 and 45: It's just general here. References are given in chapter 3 in lines approx. 70-104.

Line 48: Accepted.

Figure 4: Accepted. The image was divided into parts (c) and (d) as recommended by the reviewer.

Line 143: Accepted. Inserted reference [11] in the new numbering at line 150 and also [2] to lines 135 and reference [2] in Figure 3b in line 137.

Round 2

Reviewer 1 Report

Thanks to the authors' hard work. I am satisfied with their revision, no future review process is needed.

Reviewer 4 Report

Dear Authors,

although you didn't answer to all my comments, thank you for your effort in reviewing the paper.

Best regards.

This manuscript is a resubmission of an earlier submission. The following is a list of the peer review reports and author responses from that submission.